# Early screening of autism spectrum disorder using cry features

**Aida Khozaei**[1], **Hadi Moradi**[1,2]*, **Reshad Hosseini**[1], **Hamidreza Pouretemad**[3], **Bahareh Eskandari**[3]

**1** School of Electrical and Computer Engineering, University of Tehran, Tehran, Iran, **2** Intelligent Systems Research Institute, SKKU, Suwon, South Korea, **3** Department of Psychology, Shahid Beheshti University, Tehran, Iran

* moradih@ut.ac.ir

**Data Availability Statement:** The original and cleaned voices and their extracted features (the data set) in this research and the implementation codes of the proposed method are deposited in the following repositories: CodeOcean 10.24433/CO.

## Abstract

The increase in the number of children with autism and the importance of early autism intervention has prompted researchers to perform automatic and early autism screening. Consequently, in the present paper, a cry-based screening approach for children with Autism Spectrum Disorder (ASD) is introduced which would provide both early and automatic screening. During the study, we realized that ASD specific features are not necessarily observable in all children with ASD and in all instances collected from each child. Therefore, we proposed a new classification approach to be able to determine such features and their corresponding instances. To test the proposed approach a set of data relating to children between 18 to 53 months which had been recorded using high-quality voice recording devices and typical smartphones at various locations such as homes and daycares was studied. Then, after preprocessing, the approach was used to train a classifier, using data for 10 boys with ASD and 10 Typically Developed (TD) boys. The trained classifier was tested on the data of 14 boys and 7 girls with ASD and 14 TD boys and 7 TD girls. The sensitivity, specificity, and precision of the proposed approach for boys were 85.71%, 100%, and 92.85%, respectively. These measures were 71.42%, 100%, and 85.71% for girls, respectively. It was shown that the proposed approach outperforms the common classification methods. Furthermore, it demonstrated better results than the studies which used voice features for screening ASD. To pilot the practicality of the proposed approach for early autism screening, the trained classifier was tested on 57 participants between 10 to 18 months. These 57 participants consisted of 28 boys and 29 girls and the results were very encouraging for the use of the approach in early ASD screening.

## Introduction

Children with Autism Spectrum Disorder (ASD) are defined by their abnormal or impaired development in social interaction and communication, as well as restricted and repetitive behaviors, interests, or activities [1]. The rapid growth of ASD in the past 20 years has inspired many research efforts toward the diagnosis and rehabilitation of ASD [2–5]. In the field of

0622770.v1 Harvard Dataverse (Contains only a rar file of sounds): 10.7910/DVN/LSTBQW

**Funding:** HM received a small fund for collecting data and for diagnosing the subjects. Grant number 123 Cognitive Sciences and Technology Council of Iran cogc.ir The funders had no role in study design, data collection and analysis, decision to publish, or preparation of the manuscript.

**Competing interests:** The authors have declared that no competing interests exist.

diagnosis, there are several well-established manual methods to diagnose children over 18 months [6]. However, the practical average age of diagnosis is over 3 years due to the lack of knowledge about ASD and the lack of expertise for diagnosing autism [7, 8]. It is of the utmost importance to have early diagnosis/screening in order to provide early intervention which is more effective at the first few years of life than later on [7, 9–11]. It is shown that early intervention improves the developmental performance in children with ASD [12]. It has also been reported that early interventions would be cost saving for families and the treatment service systems [13, 14]. Consequently, there are two main questions: 1) can autism be screened earlier than 18 months to reduce the typical diagnosis or intervention age and 2) is it possible to employ intelligent methods for the screening of autism to eliminate the widespread need for experts? It should be mentioned that our goal was to answer these questions with respect to screening all children who may not have clear symptoms. The screened children should go through a diagnosis procedure to acquire confirmation and/or be cautiously worked with.

Fortunately, there are studies in the literature showing that the age of diagnosis can be lower than 18 months. For example, Thabtah and Peebles [15] reviewed several questionnaire-based approaches that may be able to screen ASD above 6 months of age. However, those approaches, like Autism Diagnostic Interview-Revised (ADI-R) [16] and Autism Diagnostic Observation Schedule (ADOS) [17] which have been clinically proven to be effective and adequate, are time-consuming instruments [15] and need trained practitioners to use them. To reduce the dependency on the human expertise needed in using such questionnaires [8], several studies proposed machine learning methods to classify children with ASD [18, 19] using questionnaires. Their goal was to automate the process and/or find an optimum subset of questions or features. For instance, Abbas et al. [20] proposed a multi-modular assessment system combined of three modules, a parent questionnaire, a clinician questionnaire, and a video assessment module. Although the authors used machine learning to automate and improve classification process, the need for human involvement still exists in order to answer questions or assess videos.

On the other hand, Emerson et. al showed that fMRI [21] can be used to predict the diagnosis of autism at the age of 2 in high-risk 6-month-old infants. Denisova and Zhao [22] used movement data from rs-fMRI from 1–2 month-old infants to predict future atypical developmental trajectories as biological features. Furthermore, Bosl, Tager-Flusberg, and Nelson [23] suggested that useful biomarkers can be extracted from EEG signals for early detection of autism. Blood-based markers [24, 25] and prenatal immune markers [26] were also proposed to diagnose ASD that can be used right after birth. Although these approaches suggest new directions towards early ASD diagnosis/screening, they are costly, require expertness and dedicated equipment, which would limit their usage. Furthermore, these methods are still in the early stages of research and require further approval. Finally, approaches which involve methods such as fMRI or EEG, are difficult to use on children, especially on children with autism who may have trouble following instructions appropriately [27], have atypical behaviors [28], or have excessive head movements [29, 30].

There are studies that used vocalization-based analysis to screen children with autism. For instance, Brisson et al. [31] showed differences in voice features between children with ASD and Typically Developing (TD) children. Several studies, like [32], used speech-related features for the screening of children older than 2. To reach the goal of early ASD screening, vocalizations of infants under 2 years of age have been investigated [33–35]. Santos et al. [33] used vocalizations, such as babbling, to screen ASD children at the age of 18 months. They collected data from 23 and 20 ASD and TD children, respectively. They reported high accuracy of around 97% which can be due to the fact that they used k-fold cross-validation without considering subject-wise hold out in order to have unseen subjects in the test fold [36]. Oller et al.

[34] proposed another vocalization-based classification method in which they included age and excluded crying. They applied the method on 106 TD children and 77 children with ASD between 16 to 48 months and reached 86% accuracy. Pokorny et al. [35] extracted eGeMAPS parameter set [37], which includes 88 acoustic parameters, in 10 month old children. This set consists of statistics calculated for 25 frequency-related, energy-related, and spectral low-level descriptors. They reached a 75% accuracy on a population of 10 TD children and 10 children with ASD.

Esposito, Hiroi, and Scattoni [38] showed that crying is a promising biomarker for the screening of ASD children. Sheinkopf et al. [39] and Orlandi et al. [40] have shown that there are differences in the cry of children with ASD compared to TD children. To the best of our knowledge, our own group's preliminary study [41] was the only research that has used cry sounds for the screening of children with ASD. We used a dataset of 5 children with ASD and 4 TD children older than two years. The accuracy of the proposed method is 96.17% using k-fold cross validation without considering subject-wise hold out, which is a shortcoming of this study. In other words, it has been overfitted to the available data and may fail to correctly classify new samples. So, a thorough examination using an unseen test set on cry features is necessary to evaluate the results. It should be noted that the data from our previous study [41] could not be used in the study presented in this paper due to the differences in data collection procedures.

In all the above studies, it was assumed that the specific sound features, distinguishing children with ASD from TD children, are common among all the ASD cases. However, this may not be the case for all the features. For instance, tiptoe walking, which is one of the repetitive behaviors of children with ASD, appears in approximately 25% of these children [42]. Consequently, in the current study, we propose a new cry-based approach for screening children with ASD. Our screening approach makes use of the assumption that all discriminative characteristics of autism may not appear in all ASD children. This assumption is in contrast with the assumption put forward in the ordinary instance-based machine learning methods, which assumes that all instances of a class include all discriminative features needed for classification. In our proposed method, at first, discriminable instances of cries, which exist in subsets of children with ASD, are found. Then it uses these instances to select features to distinguish between these ASD instances from TD instances. It should be mentioned that the final selected features, in this study, are common among our set of children with ASD between 18 to 53 months of age. These selected features support the experiential knowledge of our experts stating that the variations in the cries of children with ASD are more than TD children. This approach is different from the other approaches that either used a dataset of children with a specific age [33, 35] or used age information for classification [34]. The proposed approach has been implemented and tested on 62 participants. The results show the effectiveness of the approach with respect to accuracy, sensitivity, and specificity.

## Method

Since this study was performed on human subjects, first, it was approved by the ethics committee at Shahid Beheshti University of Medical Sciences and Health Services. All the parents of the participants were informed about the study and signed an agreement form before being included in the study.

### Participants

There were 62 participants aged between 18 and 53 months, who were divided into two groups, i.e. 31 ASD and 31 TD with 24 boys and 7 girls in each group. Since we expected to

have different vocalization characteristics for boys and girls, the training set was assembled of only boys, including 10 TD, and 10 ASD. In other words, we wanted to eliminate the gender effects on the feature extraction and model training. Unfortunately, due to the lower number of girls with ASD in the real world, not enough data for girls with ASD could be collected. Nonetheless, the model was also tested on the girls to see how it would generalize even on them.

The inclusion criteria of the ASD participants were: a) being very recently diagnosed with ASD based on DSM-5 with no other neurodevelopmental, mental, and intellectual disorder, b) having no other known medical or genetic conditions, or environmental factors, and c) not having received any treatment or medication, or having received treatment in less than a month. There were only two girls who did not fall into these criteria since they had been diagnosed more than a year before. The participants' average language development at the time of participation, which was assessed based on [43–46], was equal to children between 6 to 12 months old. The autism diagnosis procedure started with the Gilliam Autism Rating Scale-Second Edition (GARS-2) questionnaire [47] which was answered by the parents. Then the parents were interviewed, based on DSM-5, while the participants were evaluated and observed by two child clinical psychologists with Ph.D. degrees. In addition, the diagnosis of ASD was separately confirmed by at least one child psychiatrist in a different setting. It should be noted that ADOS, which is a very common diagnostic tool is not administered widely in Iran since there is no official translation of ADOS in Farsi. TD children were selected from those in an age range similar to the ASD participants from volunteer families from their homes and health centers. They had no evidence or official diagnosis of any neurological or psychological disorder at the time of recording their voices. The children with ASD were older than 20 months with the mean, standard deviation, and range of 35.6, 8.8, and 33 months respectively. The TD children were younger than 51 months with the mean, standard deviation, and range of about 30.8, 10.3, and 33 months respectively. It should be mentioned that the diagnosis of the children under 3 years was mainly based on experts' evaluation, not the GARS score. Furthermore, all TD participants under 3 years of age had a follow up study when they passed the age of 3, to make sure the initial TD assignment was correct or still valid. To do so, we used a set of expert-selected questions based on [48] to assess them through interviews with parents.

Tables 1 and 2 show the details of the participants on the training and test sets, respectively. In each table, the number of voice instances from each participant and the total duration of all its instances in seconds are shown in columns 3 and 4, respectively. The recording device category, i.e. a high-quality recorder (HQR) and typical cell phones (CP), is given in the device category column. The next two columns include GARS-2 scores and the language developmental milestone of the participants with ASD at the time of the recording. In six cases, there were no GARS score available at the time of study, demonstrated by ND (No Data). The column labeled as 'Place' shows the location of the recording which can be in homes (H), autism centers (C1, C2, and C3), and health centers (C4, C5, and C6). There was a total number of 359 samples for all children. 53.44% of the samples were from ASD participants and 46.56% were from TD participants.

Two groups of 10 TD and 10 ASD children were selected for training the classifiers such that two groups were as balanced as possible with respect to age and the recording device. Thus, each child in the TD group had a corresponding child in the ASD group around the same age. As a result of this data balancing, we obtained training participants with an age between 20 and 51 months. The mean ages in the training set were 32.7 and 35.2 months for ASD and TD participants, respectively. The standard deviations are 9 and 9.9 months with the range of 25 and 30 months for ASD and TD participants, respectively.

**Table 1. The training set data of participants.**

| | ID | Age (month) | # of instances | Total duration(sec) | Device | GARS score | Language milestone (month) | Place | Reason for crying |
|---|---|---|---|---|---|---|---|---|---|
| ASD | ASD1 | 20 | 9 | 7.8 | CP | 104 | 0–6 | C1 | Annoyed/Uncomfortable |
| | ASD2 | 24 | 3 | 1.5 | HQR | 83 | 0–6 | C2 | Unwilling |
| | ASD3 | 26 | 5 | 2.1 | HQR | 120 | 0–6 | C1 | Annoyed/Uncomfortable |
| | ASD4 | 28 | 13 | 9.1 | HQR | 121 | 0–6 | C2 | Annoyed/Uncomfortable |
| | ASD5 | 29 | 14 | 26 | HQR | 89 | 6–12 | C2 | Unwilling/Complaining |
| | ASD6 | 31 | 4 | 2.4 | HQR | 87 | 0–6 | C2 | Unwilling/Complaining |
| | ASD7 | 36 | 11 | 11 | HQR | 87 | 6–12 | C2 | Unwilling/Complaining |
| | ASD8 | 43 | 2 | 0.7 | CP | ND | ND | C2 | Unwilling |
| | ASD9 | 45 | 3 | 2.6 | CP | 72 | 6–12 | C2 | Complaining |
| | ASD10 | 45 | 4 | 3.4 | CP | ND | ND | H | Sleepy |
| TD | TD1 | 21 | 11 | 14 | HQR | NA | NA | H | Complaining |
| | TD2 | 24 | 12 | 12 | HQR | NA | NA | C4 | Scared/Unwilling |
| | TD3 | 26 | 2 | 2.3 | HQR | NA | NA | C5 | Unwilling |
| | TD4 | 28 | 6 | 13 | CP | NA | NA | C5 | Scared/Unwilling |
| | TD5 | 36 | 3 | 2.6 | CP | NA | NA | H | Unwilling/Complaining |
| | TD6 | 38 | 3 | 1.5 | HQR | NA | NA | C6 | Complaining |
| | TD7 | 41 | 3 | 2.4 | HQR | NA | NA | H | Unwilling |
| | TD8 | 43 | 3 | 2.2 | CP | NA | NA | H | Sleepy |
| | TD9 | 44 | 2 | 1.2 | CP | NA | NA | H | Complaining |
| | TD10 | 51 | 2 | 1.7 | CP | NA | NA | H | Complaining |

Although this approach was trained and tested on children older than 18 months, we tested the proposed approach on 57 participants between 10 to 18 months to investigate how it works on children under 18 months. These 57 participants consisted of 28 boys and 29 girls with the mean age of 15.2 for both and standard deviations of 2.8 and 2.9 respectively. All these participants were evaluated at a later date at the age of 3 or older, by the same follow-up procedure, using our expert-selected questionnaire. At the time of initial voice collection, 55 of these participants had no evident or diagnosed disorder. Two of them were referred to our experts due to the positive results of screening using our method. The diagnosis or concerns about the two mentioned participants, as well as the participants with any evidence of having abnormality in the developmental milestones during the follow-up procedure are summarized in Table 3. The summary of disorders given in the last column of Table 3 is based on the parental interviews and our experts' evaluation. Unfortunately, Child5, Child6, and Child7's parents did not cooperate in obtaining expert evaluation.

## Data collection and preprocessing

As mentioned earlier, the data was recorded using high-quality devices and typical smartphones. The high-quality devices were a UX560 Sony voice recorder and a Sony UX512F voice recorder. To use typical smartphones, a voice-recording and archiving application was developed and used on various types of smartphones. All voices, through the application or the high-quality recorders, were recorded in wav format, 16 bits, and with the sampling rate of 44.1 kHz. The reason for using various devices was to avoid biasing of the approach to a specific device. Similarly, the place of recording was not restricted to one place in order to make the results applicable to all places.

The parents and trained voice collectors were asked to record the voices in a quiet environment. Furthermore, they were asked to keep the recorders or smartphones about 25 cm from the participants' mouth. Despite the proposed two recommendations, there were recorded

**Table 2. The test set information.**

| | | ID | Age (month) | # of instances | Total duration(S) | Device | GARS score | Language milestone (months) | Place | Reason for crying |
|---|---|---|---|---|---|---|---|---|---|---|
| ASD | Boys | ASD11 | 28 | 12 | 7.2 | HQR | 102 | 0–6 | C2 | Unwilling/ Uncomfortable |
| | | ASD12 | 30 | 18 | 17.1 | HQR | ND | ND | C3 | Separation from mother |
| | | ASD13 | 30 | 3 | 2.9 | CP | ND | ND | H | Unwilling/Sleepy |
| | | ASD14 | 31 | 5 | 2.3 | HQR | 73 | 0–6 | C2/ H | Separation from mother/Hungriness |
| | | ASD15 | 33 | 3 | 2.5 | HQR | 91 | 0–6 | C2 | Unwilling |
| | | ASD16 | 33 | 2 | 2.5 | HQR | 104 | 0–6 | C1 | Annoyed/Uncomfortable |
| | | ASD17 | 34 | 1 | 0.6 | HQR | 91 | 0–6 | C2 | Unwilling/Complaining |
| | | ASD18 | 35 | 2 | 1.7 | HQR | 81 | ND | C1 | Annoyed/Uncomfortable |
| | | ASD19 | 37 | 1 | 0.6 | HQR | 94 | 12–18 | C2 | Unwilling/Complaining |
| | | ASD20 | 40 | 19 | 14 | HQR | 91 | 0–6 | C1 | Annoyed |
| | | ASD21 | 45 | 1 | 0.3 | HQR | 81 | 6–12 | C2 | Unwilling/Complaining |
| | | ASD22 | 48 | 2 | 1.6 | HQR | 100 | 6–12 | C2 | Annoyed/Complaining |
| | | ASD23 | 52 | 6 | 3.1 | HQR | 113 | 12–18 | C2 | Unwilling/Complaining |
| | | ASD24 | 53 | 7 | 5.2 | HQR | 78 | 6–12 | C1 | Annoyed/Uncomfortable |
| | Girls | ASD25 | 25 | 12 | 14 | HQR | 85 | 0–6 | C2 | Unwilling/Complaining |
| | | ASD26 | 26 | 5 | 2 | CP | 102 | 0–6 | C1 | Scared |
| | | ASD27 | 31 | 3 | 1.7 | HQR | 94 | 0–6 | C2 | Unwilling/Complaining |
| | | ASD28 | 32 | 2 | 1.3 | HQR | 100 | 0–6 | C2 | Unwilling/Complaining |
| | | ASD29 | 41 | 8 | 3 | HQR | 102 | 0–6 | C2 | Unwilling/Complaining |
| | | ASD30 | 45 | 2 | 1.2 | CP | ND | ND | H | Thirsty |
| | | ASD31 | 49 | 7 | 12 | CP | ND | ND | H | Unwilling/Complaining |
| TD | Boys | TD11 | 18 | 4 | 2 | HQR | NA | NA | C4 | Scared |
| | | TD12 | 18 | 7 | 5.1 | HQR | NA | NA | C4 | Scared/Unwilling |
| | | TD13 | 19 | 7 | 4.2 | HQR | NA | NA | C5 | Unwilling |
| | | TD14 | 20 | 9 | 8 | HQR | NA | NA | C5 | Unwilling/Complaining |
| | | TD15 | 21 | 4 | 1.2 | HQR | NA | NA | H | Complaining |
| | | TD16 | 24 | 3 | 2.7 | HQR | NA | NA | C5 | Scared /Unwilling |
| | | TD17 | 24 | 2 | 1.5 | HQR | NA | NA | C5 | Scared/Unwilling |
| | | TD18 | 24 | 6 | 5.1 | HQR | NA | NA | C4 | Unwilling/Complaining |
| | | TD19 | 24 | 4 | 2.4 | HQR | NA | NA | C5 | Unwilling/Complaining |
| | | TD20 | 24 | 5 | 4.2 | HQR | NA | NA | C5 | Unwilling/Complaining |
| | | TD21 | 29 | 11 | 10 | HQR | NA | NA | H | Unwilling/Complaining |
| | | TD22 | 30 | 4 | 2 | HQR | NA | NA | C5 | Scared/Unwilling |
| | | TD23 | 30 | 4 | 2 | CP | NA | NA | H | Unwilling |
| | | TD24 | 43 | 12 | 11 | HQR | NA | NA | H | Complaining |
| | Girls | TD25 | 24 | 5 | 6 | HQR | NA | NA | C4 | Unwilling/Complaining |
| | | TD26 | 25 | 2 | 4.4 | HQR | NA | NA | C5 | Scared |
| | | TD27 | 29 | 5 | 5 | HQR | NA | NA | C5 | Scared |
| | | TD28 | 33 | 2 | 2.1 | CP | NA | NA | H | Complaining |
| | | TD29 | 45 | 16 | 11 | HQR | NA | NA | H | Unwilling/Complaining |
| | | TD30 | 50 | 6 | 7 | HQR | NA | NA | H | Complaining |
| | | TD31 | 51 | 2 | 0.7 | CP | NA | NA | H | Unwilling |

voices where the recommendations were not followed and did not have the required quality. Consequently, those recordings were eliminated from the study. Also, all the cry sounds which were due to pain, had been removed from the study since they were similar between the TD and ASD groups.

**Table 3. The participants with an abnormality in the follow-up.**

| ID | Gender | Age (in months) | | Disorder |
|---|---|---|---|---|
| | | at recording time | at following-up time | |
| Child1 | M | 11 | 11 | Developmental delay[a], signs of genetic diseases |
| Child2 | M | 17 | 17 | UNDD[b] |
| Child3 | M | 12 | 40 | ASD[b] |
| Child4 | M | 12 | 36 | Sensory processing disorder[c], several ADHD symptoms[b] |
| Child5 | M | 18 | 40 | Language delay |
| Child6 | M | 15 | 46 | Developmental delay symptoms |
| Child7 | M | 12 | 43 | Developmental delay symptoms |

UNDD, Unspecified Neurodevelopmental Disorder.

[a] Clinical observation by our expert based on [48].

[b] Clinical observation by our expert based on [1]

[c] Clinical observation by our expert based on [49].

After data collection, there was a preprocessing phase in which only pure crying parts of the recordings, with no other types of vocalization, were selected. To explain more, the parts of cry sounds which were accompanied by screaming, saying words/other vocalizations, or that occurred with closed/non-empty mouth were eliminated. All segmentations and eliminations were done manually using Sound Forge Pro 11.0. From the selected cries, the beginning and the end, which contained voice rises and fades, were removed in order to just keep the steady parts of the cries; this prevents having too much variation in the voice which can lead to unsuitable statistics. Also, the uvular/guttural parts of the cries were removed. The reason for this was that we believe these parts distort the feature values of the steady parts of a voice. Each remaining continuous segment of the cries was considered and used as a sample (instance) in this study. Finally, since the basic voice features were extracted from 20 milliseconds frames [50], to generate statistical features of the basic features, the minimum length of the cry segments were set to 15 frames, i.e. 300 milliseconds. Thus, any cry samples below 300 milliseconds were eliminated from the study. In this study, the final prepared samples were between 320 milliseconds to 3 seconds.

### Feature extraction

Previous studies working on voice features for discriminating ASD children used different sets of features. These methods share several common features like F0, i.e. the fundamental frequency of a voice, and Mel-Frequency Cepstral Coefficients (MFCC), i.e. coefficients which represent the short-term power spectrum of a sound [51]. F0 has been one of the most common features used [31, 32, 39]. However, since age is an important factor affecting F0 [52], this feature is useful when participants have a similar age. On the other hand, MFCC coefficients and several related statistical values have been reported to be useful features in several studies [35, 41, 53]. Considering the useful features reported in previous studies and the specifications of the current study, several features were selected to be used in this work that are explained in the following.

In this study, each instance was divided into 20 milliseconds frames, to extract basic voice features. We used several features proposed by Motlagh, Moradi, and Pouretemad [41] and by Belalcázar-Bolaños et al. [54]. The features used by Motlagh, Moradi, and Pouretemad [41] include certain statistics like mean and covariance of the frame-wise basic features, such as MFCC coefficients, over a voice segment. They also used the mean and variance of frame-wise temporal derivative [55, 56] of the basic features. The frame-wise temporal derivative means

the difference between two consecutive frames, which in a sense is the rate of change of a feature value in one frame step. We modified the spectral flatness features by including the range of 125–250 Hz beside the 250–500 Hz range. This range was added to cover a wider frequency range than the normal children frequency range, which showed to be necessary in the process of feature extraction and selection. Each range is divided into 4 octaves and the spectral flatness is computed for those octaves.

We removed all uninformative and noisy features of the set which are explained in the following. The mean of frame-wise temporal derivative of the basic features is removed because it is not a meaningful feature and is equal to taking the difference between the value of the last and the first frames. There are means of the features related to the energy, such as the audio power, total loudness, SONE, and the first coefficient of MFCC, that were removed to make the classifier independent of the loudness/power in children's voices. Zero crossing rate (ZCR) was omitted too, due to its dependency on the noise in the environment.

The second set of features used in this study was from Belalcázar-Bolaños et al. [54] because it has phonation features, like jitter and shimmer. Jitter and shimmer, which have been reported to be discriminative for ASD, are linked to perceptions of breathiness, hoarseness, and roughness [57]. Other features used from Belalcázar-Bolaños et al. [54] include glottal features related to vocal quality and the closing velocity of the vocal folds [33]. The mean of logarithmic energy feature was omitted for the same reason as other energy-related features. A summary of the features, added to or removed from the sets by [41] and [54], is presented in Table 4.

## The proposed subset instance classifier

To explain the proposed classifier, it was assumed that there is a target group of participants that we want to distinguish from the rest of the participants, called the rest. Furthermore, each participant in the target group may have several instances that may be used to distinguish the target group from the rest. Fig 1A shows a situation in which all instances of all participants of the target group are differentiable using common classifiers that we call Whole Set Instance (WSI) classifiers. In this figure, the circles represent our target group and the triangles represent the rest. The color coding is used to differentiate between the instances of each participant in each group. In contrast to the situation in Fig 1A, in Fig 1B the target group cannot easily be distinguished from the rest. In such a situation, there are instances of two participants in the target group, i.e. the red and brown circles that are not easily separable from the instances in the rest (Case 1). Furthermore, there is a participant with no instances, i.e. the orange circles, easily

**Table 4. The features and statistics which were added or removed to the two feature sets.**

| | Feature | removing/adding | Reason |
|---|---|---|---|
| Second set | logarithmic energy | Mean statistic is removed | Classification dependency on loudness/power of cries |
| First set | Audio power | | |
| | Total loudness | | |
| | SONE | | |
| | First MFCC coefficient | | |
| | ZCR | The basic feature is removed | The feature's dependency on environmental noise |
| | All basic features applicable | mean of frame-wise temporal derivative of the basic features is removed | No meaning for the feature |
| | MFCC | Coefficients of 14–24 are added | Having higher-order coefficients for vocal cords information as well as vocal tract |
| | Spectral flatness | A range of 125–250 Hz is added | Covering the low-frequency range of human voice |

separable from the rest (Case 2). An example of Case 1 is tiptoe walking in children with ASD, which is common in about 25% of these children [42] who do it most of the time. An example of Case 2 is children with ASD who do not tiptoe walk. In other words, there are children with ASD who cannot be distinguished from TD children using the tiptoe walking behavior factor.

Applying any WSI classifier may fail for the data type shown in Fig 1B. Consequently, we proposed SubSet Instance (SSI) classifier that first finds differentiable instances and then trains a classifier on these instances. As an example, the proposed SSI classifier first tries to find the circles on the left of the line in Fig 1B, using a clustering method. Then, it uses these circles, as *exclusive instances* having a specific feature common in a subset of the target group, to train a classifier separating a subset of the target group.

The steps of common WSI classifiers are shown in Fig 2A. The steps of our proposed SSI classifier are shown in Fig 2B. In the SSI classification approach, after the feature extraction and clustering steps, for each cluster, a classifier is trained to separate its exclusive instances from the instances of the rest of the participants. In the testing phase, any participant with only one instance classified in the target group (positive instance), is classified as a target group's participant. The pseudo-code for the proposed approach is given in Algorithms 1 and 2.

## Algorithm 1. Training SSI classifiers

```
T: set of all target group instances
R: set of all the rest instances
F: set of all classifiers
ρ: threshold for the number of samples in a cluster
s: the number of minimum samples needed in a cluster to be able to
train a classifier for it
Cⱼ: The jᵗʰ cluster
n: number of clusters
F = ∅
1: While ∃j |Cj| > ρ; while there is a cluster bigger than a threshold
or n = 1
2:   n = n + 1; increase the number of clusters
3:   Cluster the T + R into n clusters Cⱼ, j = 1,...,n
4:   EC = {Cⱼ ⊂ T}; the set of clusters of only exclusive instances,
i.e. exclusive clusters
```

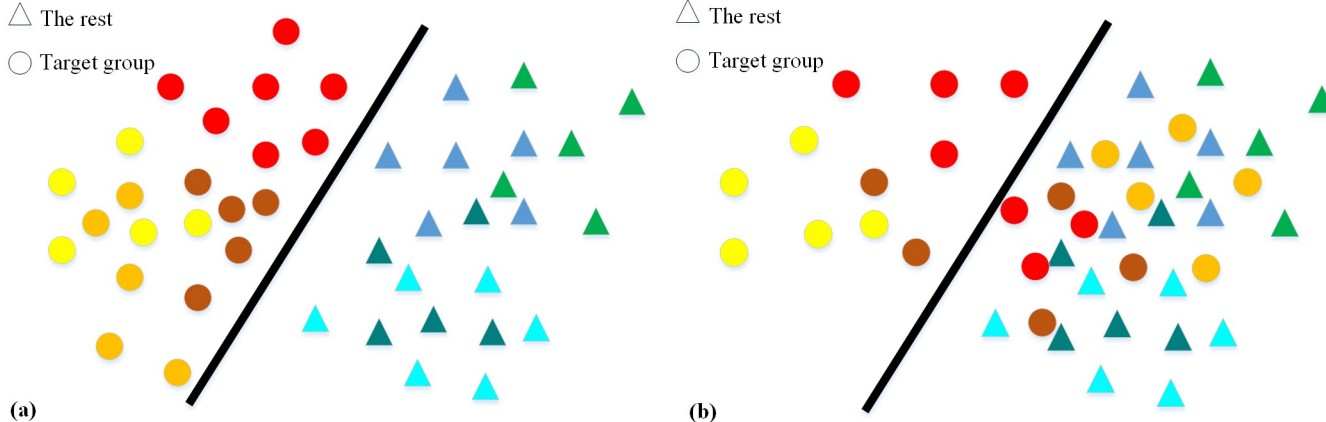

**(a)** **(b)**

**Fig 1. Two different hypothetical types of two-dimensional data of the target group and the rest.** The instances shown by the warm-colored circles and the cool-colored triangles are for the target group and the rest, respectively. All instances belonging to a participant have the same color. In (a), all the target group participants' instances are distinguishable using a classifier. In (b), only some instances of the target group participants are separable from the other instances by a classifier.

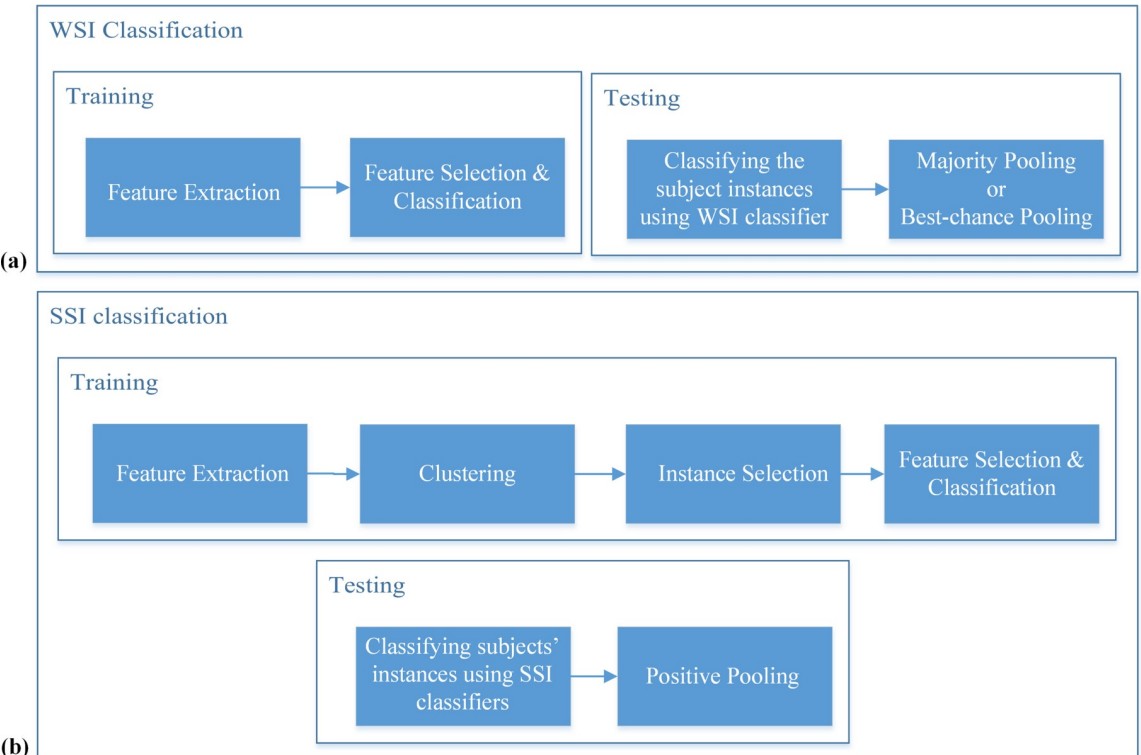

**Fig 2. An overall view of WSI and SSI methods.** (a) In WSI method, after feature extraction, a classifier is trained on all instances and majority pooling (MP) is usually used in the testing phase. In this study Best-chance threshold Pooling (BP), which is a threshold-based pooling with the threshold giving the best accuracy on the test set, is also used to give the best chance to WSI classifier. (b) In the proposed SSI classifier, after feature extraction, clustering is applied to find and select exclusive instances containing instances of the target group participants only. Then classifiers are trained using exclusive instances, and a participant is classified in the target group in the testing phase if any classifier detects a positive instance for it.

```
5:    If EC ≠ ∅; check if there is any exclusive cluster
6:       For all Cⱼ in EC with |Cj| > s
7:          Train a classifier using positive labels c ϵ Cⱼ and negative
labels r ϵ R
8:          Add the classifier to F
9:       T = T − ∑    Cⱼ; remove the instances of the exclusive clusters from
              Cⱼ⊂EC
target group instances
10:       n = 1; set 1 to re-start clustering in two groups on the remain-
ing instances
```

## Algorithm 2. Testing SSI classifiers

```
F: set of trained classifiers
A: set of subject instances
1:  For all instances a of A
2:    P = {a ∈ A|∃f, classifies a as positive instance}
3:    If P ≠ ∅
4:      The participant is from the target group
5:    Else
6:      The participant is from the rest
```

In the proposed training algorithm of the SSI approach, the goal is to find clusters containing the ASD instances only. Then a classifier is trained using the instances of these clusters and

added to a list of all trained classifiers (lines 7 and 8 of Algorithm 1). As shown in the loop of the algorithm, starting at line 1, the data is clustered starting with two clusters. Then the number of clusters is increased until a cluster, containing only the target group instances, emerges. The exclusive instances in such a cluster are removed from the set of all target group's instances, and the loop is restarted. Before restarting the loop, if the number of instances in this cluster is more than a threshold, a new classifier using these instances is trained and this classifier is added to the set of all trained classifiers. The loop stops when the number of samples in each cluster is less than a threshold.

For testing the participants, using the trained classifiers, all the instances of each participant are classified one by one using all the trained classifiers (line 2 of Algorithm 2). A subject would be classified in the target group if at least one of its instances is classified in the target group at least by one of the classifiers (lines 3 and 4 of Algorithm 2). Otherwise, if there is no instance classified among the target group, the participant is classified as the rest (lines 5 and 6).

## Details of the implementations

The classifiers were implemented in Python using scikit-learn library.

**WSI classifiers.**   We have tested several common WSI classifiers, but we report only the result of SVM with RBF kernel and with no feature selection, which gives the best average accuracy. It should be noted that several feature selection approaches, like L1-SVM and backward elimination, were tested but they only reduced the accuracy. We used group 5-fold cross-validation for tuning hyper-parameters. Group K-fold means that all instances of each participant are placed in only one of the folds. This prevents having the same participant's instances in the train and validation folds simultaneously. In each fold, there were two ASD and two TD participants. It should be mentioned that before applying the algorithms, we balanced the number of instances of the two groups using upsampling.

Two approaches were exploited to combine the decisions on different samples of a participant in the WSI approach. The first approach was majority pooling which classifies a participant as ASD, if the number of instances classified as ASD are more than 50 percent of all instances. The second approach was threshold-based pooling which is similar to the first approach except that a threshold other than 50 is used.

**SSI classifiers.**   Before applying the algorithm, we balanced the number of instances of the two groups by upsampling. The threshold for the minimum number of samples, needed in a cluster, to be able to train a classifier is set to 10. It should be mentioned that agglomerative clustering and decision tree are the methods used for clustering and classification parts of Algorithm 1, respectively.

**Training the SSI classifiers.**   After running Algorithm 1 on our data, two exclusive clusters with enough instances, i.e. at least 10 instances in our study, were found. Then two classifiers were trained corresponding to each cluster. One of these exclusive clusters had 11 instances from 4 ASD participants (Table 1). These 11 instances consisted of 6 out of 9 instances of ASD1, 2 out of 4 instances of ASD10, 1 out of 2 instances of ASD8, and 2 out of 4 instances of ASD6. As explained in the algorithm, for each cluster, a decision tree classifier was trained using the ASD instances in the cluster versus all TD instances. Interestingly, only one feature was enough to discriminate instances in the cluster from all TD instances. Among those features that can discriminate the cluster's instances, we selected the Variance of Framewise Temporal Derivative (VFTD) of the 7th MFCC coefficient as the feature which can discriminate more ASD participants from the set of all participants with a simple threshold. The classifier obtained by setting a threshold based on this feature was the first classifier. This feature supports our expert's report regarding the higher variations in the cry sounds of ASD

children than TD children. From 10 ASD children, 8 of them can be discriminated using this feature. For each participant, the number of instances found by this classifier is shown in the 2nd column of Table 5.

After excluding the ASD samples from the first classifier, the second classifier was trained based on the second exclusive cluster. This cluster included all instances of participant ASD4. The only feature used for classifying this cluster was VFTD of the 6th SONE coefficient. SONE is a unit of loudness which is a subjective perception of sound pressure [58]. Having higher VFTD of the 6th SONE coefficient confirms the experiential knowledge of our experts mentioned before. Among all the ASD participants, eight had instances with VFTD of the 6th SONE higher than a threshold (Shown in the 3rd column of Table 5). The results of classification based on these two features are depicted in Fig 3. As mentioned in the proposed method section, the participants with at least one instance classified into this cluster would be considered as a participant with ASD.

## Results

In this part, the performance of our proposed SSI classifier against a common WSI classifier is evaluated on our test set of ASD and TD participants. Each participant has multiple instances which are cleaned using the criteria explained in the data collection and preprocessing section. The participants who had at least one accepted instance were used in the training and testing phases, which are shown in Tables 1 and 2.

The output of the SSI approach was two classifiers, each of them works by setting a threshold based on a feature. The number of instances of ASD participants in the training set, correctly detected by the first and the second classifiers, are shown in the second and third columns of Table 5, respectively. On the other hand, the best-resulting classifier for the WSI approach was Radial Basis Function-Support Vector Machine (RBF-SVM) [59].

The classification results on the test set for different classifiers are shown in Table 6. The portion of each participant's instances, correctly classified by each classifier, is written as a percentage under the name of the classifier. The decision made by the WSI and SSI classifiers for each participant is shown by ASD or TD. To classify each subject using the WSI classifier, the Majority Pooling (MP) and the Best-chance threshold Pooling (BP) approaches were used. BP is a threshold-based pooling with the threshold giving the best accuracy on the test set for male participants. For the boys, MP has specificity, sensitivity, and precision equal to 100%, 35.71%, and 67.85%, respectively. On the other hand, BP leads to specificity, sensitivity, and precision equal to 85.71%, 71.42%, and 78.57%, respectively. The threshold

**Table 5. The number of instances of each participant in the training set that are classified as ASD using each trained SSI classifier.**

| ID | First SSI classifier | Second SSI classifier |
|---|---|---|
| ASD1 | 8 | 3 |
| ASD2 | 1 | 2 |
| ASD3 | 3 | 1 |
| ASD4 | 10 | 9 |
| ASD5 | 0 | 0 |
| ASD6 | 1 | 3 |
| ASD7 | 1 | 0 |
| ASD8 | 1 | 2 |
| ASD9 | 0 | 1 |
| ASD10 | 2 | 4 |

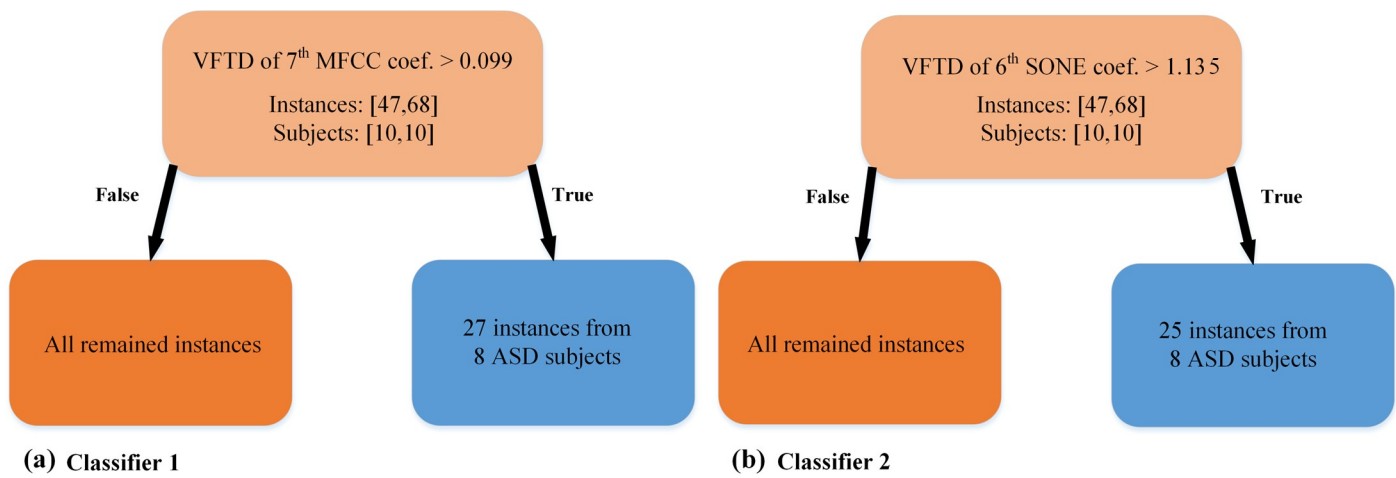

**Fig 3. Two classifiers trained on the two exclusive clusters found during the SSI classifier training phase.** (a) The Variance of Frame-wise Temporal Derivative (VFTD) of the 7th MFCC coefficient separates 27 instances of 8 ASD subjects from all TD instances of the training set. (b) VFTD of the 6th SONE coefficient separates 17 instances of 7 ASD participants from all TD instances of the training set.

for BP was set to 20% that means if 20% of instances of a participant were classified as ASD instance, the participant was classified as having ASD. The results of the percentage of instances correctly classified by the two classifiers in the SSI approach are shown as $C_1$ (the first SSI classifier) and $C_2$ (the second SSI classifier) in Table 6. The aggregated result of the decisions by $C_1$ and $C_2$ makes the final decision of the SSI classifier which is shown in the decision column, under the SSI classification section. The achieved specificity, sensitivity, and precision using the proposed method for the boys are 100%, 85.71%, and 92.85%, respectively.

To further show the applicability of the proposed approach to girls, we applied the boys' trained classifiers on the test set of the girls. The results are shown in the last row of Table 6 which show that the MP approach has specificity, sensitivity, and precision equal to 100%, 71.42%, and 85.71%, respectively. Furthermore, the BP approach gives specificity, sensitivity, and precision all equal to 85.71%, respectively. The results of the proposed SSI classifier is 100% specificity, 71.42% sensitivity, and 85.71% precision.

A two-dimensional scatter plot of the two features, used in $C_1$ and $C_2$ classifiers, are shown in Fig 4. As can be seen in this figure, the instances of a participant with ASD are scattered in the area containing instances of both TD and ASD participants. Nevertheless, there are instances for this participant uniquely distinguishable using the selected two features.

We compared the results of our proposed method with that of the only method available in the literature which was trained using only cry features [41] based on our data. The results (Table 7) show the superiority of our method, compared to the previously proposed method.

### Investigating the trained classifier on participants under 18 months

The SSI classifier which was trained using the training set in Table 1 was also tested on the data of children younger than 18 months. From 57 participants under 18 months, two boys (Child1 and Child2 in Table 3) were classified as ASD by the mentioned trained classifier. These participants were referred to our experts for diagnosis. These two were suspected of having neurodevelopmental problems. All other boys were classified as TD. However, among them, Child3 was diagnosed with ASD at the age of 2. Also, Child4 showed

**Table 6. The results of classifiers on the instances of each participant in the test set.**

| | ID | TD children WSI classification SVM | Dec. MP | Dec. BP | SSI classification C₁ | C₂ | Dec. | ID | Children with ASD WSI classification SVM | Dec. MP | Dec. BP | SSI classification C₁ | C₂ | Dec. |
|---|---|---|---|---|---|---|---|---|---|---|---|---|---|---|
| Boys | TD11 | 100 | TD | TD | 100 | 100 | TD | ASD11 | 50 | ASD | ASD | 17 | 50 | ASD |
| | TD12 | 100 | TD | TD | 100 | 100 | TD | ASD12 | 33 | TD | ASD | 11 | 28 | ASD |
| | TD13 | 100 | TD | TD | 100 | 100 | TD | ASD13 | 33 | TD | ASD | 33 | 0 | ASD |
| | TD14 | 100 | TD | TD | 100 | 100 | TD | ASD14 | 20 | TD | ASD | 20 | 20 | ASD |
| | TD15 | 100 | TD | TD | 100 | 100 | TD | ASD15 | 0 | TD | TD | 0 | 40 | ASD |
| | TD16 | 100 | TD | TD | 100 | 100 | TD | ASD16 | 50 | ASD | ASD | 100 | 0 | ASD |
| | TD17 | 100 | TD | TD | 100 | 100 | TD | ASD17 | 0 | TD | TD | 0 | 100 | ASD |
| | TD18 | 83 | TD | TD | 100 | 100 | TD | ASD18 | 50 | ASD | ASD | 50 | 50 | ASD |
| | TD19 | 100 | TD | TD | 100 | 100 | TD | ASD19 | 0 | TD | TD | 0 | 0 | TD |
| | TD20 | 80 | TD | ASD | 100 | 100 | TD | ASD20 | 42 | TD | ASD | 42 | 16 | ASD |
| | TD21 | 100 | TD | TD | 100 | 100 | TD | ASD21 | 100 | ASD | ASD | 0 | 0 | TD |
| | TD22 | 100 | TD | TD | 100 | 100 | TD | ASD22 | 0 | TD | TD | 0 | 50 | ASD |
| | TD23 | 75 | TD | ASD | 100 | 100 | TD | ASD23 | 33 | TD | ASD | 33 | 17 | ASD |
| | TD24 | 92 | TD | TD | 100 | 100 | TD | ASD24 | 86 | ASD | ASD | 86 | 86 | ASD |
| | Acc. % | | 100 | 85.71 | | | 100 | | | 35.71 | 71.42 | | | 85.71 |
| Girls | TD25 | 100 | TD | TD | 100 | 100 | TD | ASD25 | 42 | TD | ASD | 17 | 0 | ASD |
| | TD26 | 100 | TD | TD | 100 | 100 | TD | ASD26 | 60 | ASD | ASD | 60 | 20 | ASD |
| | TD27 | 100 | TD | TD | 100 | 100 | TD | ASD27 | 50 | ASD | ASD | 0 | 0 | TD |
| | TD28 | 100 | TD | TD | 100 | 100 | TD | ASD28 | 100 | ASD | ASD | 0 | 50 | ASD |
| | TD29 | 100 | TD | TD | 100 | 100 | TD | ASD29 | 62 | ASD | ASD | 50 | 50 | ASD |
| | TD30 | 67 | TD | ASD | 100 | 100 | TD | ASD30 | 100 | ASD | ASD | 50 | 50 | ASD |
| | TD31 | 100 | TD | TD | 100 | 100 | TD | ASD31 | 0 | TD | TD | 0 | 0 | TD |
| | Acc. % | | 100 | 85.71 | | | 100 | | | 71.42 | 85.71 | | | 71.42 |

Each classifier result on a participant's instances is reported as a percentage.

Dec., Decision; MP, Majority Pooling; BC, Best-chance threshold Pooling; C1, Classifier1; C2, Classifier2; Acc., Accuracy.

symptoms of having ADHD and sensory processing disorder at the age of 3. Three other children had symptoms which suggested that they are not TD children. Two of the girls who were 18 months old were classified as ASD, using the trained classifier. The other girls were classified as TD. The results of testing the trained SSI classifier on this data set are summarized in Table 8.

The original and cleaned voices and their extracted features (the data set) in this research and the implementation codes of the proposed method are deposited in the following repositories:

CodeOcean

10.24433/CO.0622770.v1

Harvard Dataverse (Contains only a rar file of sounds):

10.7910/DVN/LSTBQW

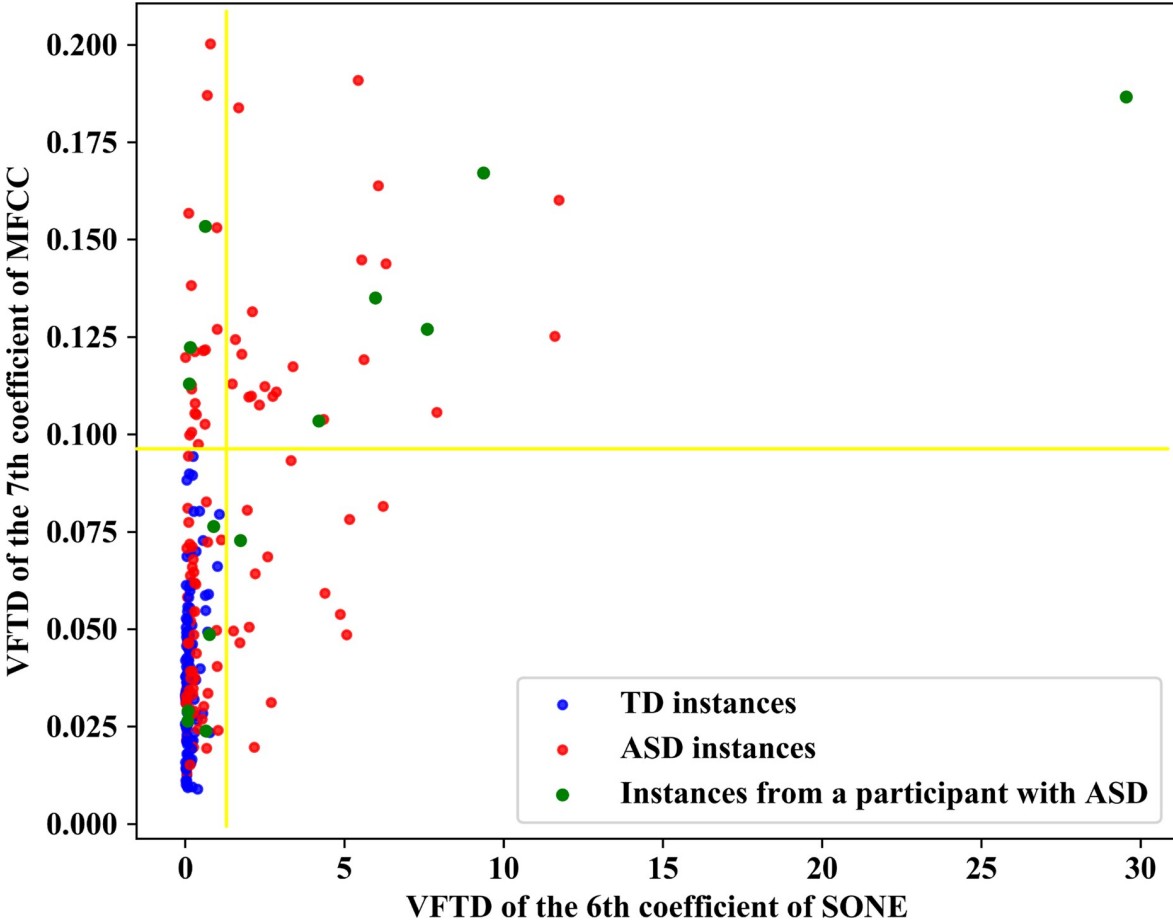

**Fig 4. Instances of several ASD and TD participants scattered in the space of two features given by the proposed SSI method.** The instances of a chosen ASD participant are illustrated in green to show that a participant may have instances in the area common with TD instances besides those two areas separated by the selected thresholds as ASD. The mentioned ASD participant (with green instances) is tagged as ASD, due to having at least one instance with the greater value than at least one of the thresholds on the two features.

## Discussion and conclusion

In this paper, we presented a novel cry-based screening method to distinguish between children with autism and typically developing children. In the proposed method, groups of children with autism who have specific features in their cry sounds can be determined. This method is based on a new classification approach called SubSet Instance (SSI) classifier. An appealing property of the proposed SSI classifier, in the case of voice-based autism screening, is its high specificity such that a normal child can be detected with no error. We applied the proposed method on a group of participants consisting of 24 boys with ASD between 20 and

**Table 7. Comparison of the results on the test set using the two methods; SSI approach and a baseline approach.**

|  |  | Sensitivity | Specificity | Precision |
|---|---|---|---|---|
| Boys | SSI | 85.71% | 100% | 92.85% |
|  | Baseline | 50.58% | 81% | 65% |
| Girls | SSI | 71.42% | 100% | 85.71% |
|  | Baseline | 21% | 86.48% | 53% |

**Table 8. Classification of the participants under 18 months using our trained SSI classifier.**

| | Boys | | | Girls | | |
|---|---|---|---|---|---|---|
| | ASD | TD | Others[a] | ASD | TD | Others[a] |
| Classified as ASD | 0 | 0 | 2 | 0 | 1 | 0 |
| Classified as TD | 1 | 22 | 4 | 0 | 27 | 0 |

[a] Other developmental or mental disorders

53 months of age and 24 TD boys between 18 and 51 months of age. The two features, found in this study, were used to train a classifier on 10 boys with ASD and 10 TD boys. Then, the classifier was used to distinguish 14 boys with ASD from 14 TD boys, reaching 92.8% accuracy. Due to the fact that girls are less likely to have autism and consequently, it is harder to collect enough data from girls than boys, the number of girls with ASD was not sufficient to train a separate classifier for this gender. It should be noted that we tested the trained system on 7 girls with ASD and 7 TD girls. It was seen that the trained classifier can screen girls with 7% lower accuracy than boys of the test set. In other words, it seems that gender differences should be considered in the training of the system. In testing the data from participants under 18 months, one TD girl was classified as ASD which was not the case for any TD children of the male counterparts. This result also confirms the aforementioned point about the gender effect. However, in future work, we would try to collect more data on girls to be able to train a system to accurately screen girls. Furthermore, we would also try to train a single classifier for boys and girls to determine whether it can be used for both of them.

It should be mentioned that our training and test data were completely separate, to make the trained model more general. The features found in this study are applicable in the age range of our participants from 18 to 53 months. This is in contrast to other approaches that either used a dataset of children with a specific age [33, 35] or used age information for classification [34]. Due to the age invariant features found in this study, it can be claimed that there are markers in the voices of children with ASD that are sustained at least in a range of ages.

The two discriminative features, found in this study, were a coefficient of MFCC and a SONE coefficient. MFCC and SONE are related to the power spectrum of a speech signal. SONE measures loudness in specific Bark bands [56]. On the other hand, MFCC, which is the inverse DFT of log-spectrum in the Mel scale, is related to the timbre of the voice [60]. Therefore, MFCC and SONE can be interpreted to be related to the timbre and loudness of a tone. Furthermore, based on the feedback from our experts, there is unpredictability in the crying sound of children with autism which is not the case for TD children. Consequently, we used the variance of temporal difference as a feature suitable for screening children with autism. This is due to the fact that if a signal is constant or changes linearly over time, the variance of temporal difference is zero. Therefore, the variance of temporal difference can be seen as the amount of ambiguity or unpredictability of a sound. On the other hand, the heightened variability in the two features, found in this study, for children with ASD is significant due to the reports from other studies [22, 61] which shows increased biological signals variability in children with ASD and infants at high risk for autism in comparison with TD children. These features are statistical features of the cry instances that hold constant, at least, across an age range studied in this research.

To the best of our knowledge, [34] and [35] were the only studies on screening children with autism using voice features on children younger than 2 years of age. Our proposed method has higher precision than these two, i.e. 6% more than [34] and 17% more than [35], using only cry features. The use of cry features as suitable biomarkers for autism screening matches the claims in [38].

In the present study only children with ASD and TD children were tested. Other developmental disorders or health issues were not tested to see how children with such disorders would be classified using the proposed method which can decrease the specificity of 100%. However, this approach is proposed to be used as a screening tool and the final diagnosis should be done under experts' supervision. So, this approach can be applied as a general screener of autism spectrum disorder.

The trained classifier was also tested on 57 participants between 10 to 18 months of age. The classifier screened two boys from the rest, i.e. Child1 and Child2 (Table 3). Child1 showed evidences of genetic disease and was diagnosed with developmental delay and Child2 received UNDD classification by our experts. This suggests that a) the system can be used for children under 2 years of age, and b) it may be able to distinguish other neurodevelopmental disorders. On the other hand, there were 5 boys, i.e. Child3 to Child7 (Table 3), who had no evidence of mental or developmental disorders at the time of their recording. At the same time, our approach did not distinguish them as children with ASD either. However, when they were older than 3 years, they showed symptoms of neurodevelopmental disorders. Out of these children, we could manage to collect new recordings from Child3 and Child4 that were classified as children with ASD using our approach. Unfortunately, Child5, Child6, and Child7 did not cooperate and could not be evaluated by an expert to validate the results of our expert-selected questionnaire. Furthermore, the parents refused to cooperate send us their children's recent cry sounds.

The result of studying these 57 children under the age of 18 months may suggest that: a) there could be symptoms in the crying sounds of children with neurodevelopmental disorders under 18 months (Child1 and Child2), b) the approach may not be able to screen a participant with neurodevelopmental disorders under the age of 18 months due to the possibility that: 1) the participant was among those children with neurodevelopmental disorders who do not have our proposed specific features in their crying sounds, 2) the participant's recorded cry samples did not include our specific features, and/or 3) neurodevelopmental disorders and their features had not been developed in the child at the time of initial recording. The reason behind not classifying Child3 and Child4, as children with ASD under the age of 18, could be b.2 or b.3. To clearly determine any reason behind this phenomena, a further investigation is needed.

We believe that this approach can be used to perform early autism screening under 18 months of age. Thus, in the future, we need to collect data and test the approach on more data of children under 18 months to validate these results with more confidence.

We have to further check the proposed approach and the extracted features on other neurodevelopmental disorders, such as ADHD, to evaluate the capability of the approach to distinguish the children with these disorders from TD children.

Furthermore, without comparing the cry sounds of children with ASD to those without ASD but another disorder, we do not really know if these findings are specific to autism or to general atypical brain developments. Thus, we should collect cry sounds of children with other neurodevelopmental disorders and compare voices of children with ASD to voices of children with other neurodevelopmental disorders to see if these features would be able to separate them or not.

It has been demonstrated that crying consists of intricate motor activities [62]. On the other hand, it has been shown that children with ASD have problems in the motor domain and in coordination of their motor capabilities with other modalities [63]. Consequently, it is possible that the extracted features in the crying sounds of children with ASD come from this deficiency/problem in the motor domain which requires further investigations.

Finally, automating the preprocessing part is a technical issue that should be addressed if it is deemed necessary that the cry-based screening be fully automated. This is important since

such a screening system can be deployed in systems such as Amazon Alexa [64] to automatically screen problematic cry sounds.

## Acknowledgments

We would like to thank the Center for Treatment of Autism Disorder (CTAD) and its members for supporting this study. We would also like to thank all the families who helped this research by taking the time to collect the cry sounds of their children. The authors would also like to express their gratitude to Prof. H. Sameti from Sharif University of Technology for his valuable and constructive feedbacks on the data collection and voice processing.

## Author Contributions

**Conceptualization:** Aida Khozaei, Hadi Moradi, Reshad Hosseini, Hamidreza Pouretemad, Bahareh Eskandari.

**Data curation:** Aida Khozaei.

**Formal analysis:** Aida Khozaei.

**Funding acquisition:** Hadi Moradi.

**Investigation:** Hadi Moradi.

**Methodology:** Aida Khozaei.

**Project administration:** Hadi Moradi.

**Software:** Aida Khozaei.

**Supervision:** Hadi Moradi.

**Validation:** Aida Khozaei.

**Visualization:** Aida Khozaei.

**Writing – original draft:** Aida Khozaei, Hadi Moradi, Reshad Hosseini.

**Writing – review & editing:** Aida Khozaei, Hadi Moradi, Reshad Hosseini, Hamidreza Pouretemad, Bahareh Eskandari.

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
