## [Decision Letter · Decision Letter 0]

16 Apr 2020

PONE-D-19-32813

Early screening of autism spectrum disorder using cry features

PLOS ONE

Dear Professor Moradi,

Thank you for submitting your manuscript to PLOS ONE. After careful consideration, we feel that it has merit but does not fully meet PLOS ONE’s publication criteria as it currently stands. Therefore, we invite you to submit a revised version of the manuscript that addresses the points raised during the review process.

We would appreciate receiving your revised manuscript by May 31 2020 11:59PM. To enhance the reproducibility of your results, we recommend that if applicable you deposit your laboratory protocols in protocols.io, where a protocol can be assigned its own identifier (DOI) such that it can be cited independently in the future. For instructions see: http://journals.plos.org/plosone/s/submission-guidelines#loc-laboratory-protocols

We look forward to receiving your revised manuscript.

Kind regards,

Zhishun Wang, Ph.D.

Academic Editor

PLOS ONE

Journal Requirements:

3. We note from your ethics statement that 'The study has been approved by the ethics committee at Shahid Beheshti University of

Medical Sciences and Health Services. All the parents of the subjects were informed about the study and signed an agreement to be included in the study.'

Please include this information in the methods section of the manuscript.

6. Your ethics statement must appear in the Methods section of your manuscript. If your ethics statement is written in any section besides the Methods, please move it to the Methods section and delete it from any other section. Please also ensure that your ethics statement is included in your manuscript, as the ethics section of your online submission will not be published alongside your manuscript.

Reviewers' comments:

Reviewer's Responses to Questions

**Comments to the Author**

1. Is the manuscript technically sound, and do the data support the conclusions?

Reviewer #1: No

Reviewer #2: Partly

Reviewer #3: No

2. Has the statistical analysis been performed appropriately and rigorously? 

Reviewer #1: Yes

Reviewer #2: I Don't Know

Reviewer #3: No

3. Have the authors made all data underlying the findings in their manuscript fully available?

Reviewer #1: No

Reviewer #2: Yes

Reviewer #3: Yes

4. Is the manuscript presented in an intelligible fashion and written in standard English?

Reviewer #1: Yes

Reviewer #2: No

Reviewer #3: No

5. Review Comments to the Author

Reviewer #1: The research reported in this article follows a recent trend towards the development of objective and easily implemented biomarkers that can be used to distinguish individuals with vs. without autism and in some cases, identify autism in the first year of life. In this study machine learning and other AI-based tools were applied to children’s cries and were found to have high sensitivity and specificity in distinguishing children with and without autism.

The authors are to be applauded for undertaking a problem of such great public health importance. However, there are several limitations of the work that detracts from its overall impact. These are listed below

a) Autism is a notoriously heterogenous disorder, and as a result, the sample size used in this work is likely not representative of autism generally. If the algorithms developed by this team are as robust as they purport, then they should be applied to publicly available data on a much larger scale.

b) No mention is made of whether the children in this study had IQ or language in the typical range or whether some had an underlying intellectual disability or speech/language problem, both of which could contribute to atypical cries having nothing to do with autism per se

c) Similarly, no mention is made of whether any of the children contributing data had an underlying genetic disorder, which again could lead to atypical (and distinct) cries.

d) Without comparing the cries of children with ASD to those without ASD but another disorder, we don’t really know if these findings are specific to autism or to atypical brain development generally.

e) Little attention is seemingly paid to the context for children’s cries, and this would seem important to understand; thus, were the cries related to distress, to pain, to hunger, to fatigue, etc?

f) I was a little confused about the gender difference the authors report. My sense is that the training set only involved male cries but the test set involved both males and females? If that is correct, I don’t know why a female training set was not developed.

g) Finally, in several places in the text the authors appear to conflate screening for autism with early identification of autism; these are two very different problems.

Reviewer #2: Thank you for the opportunity to review the manuscript “Early screening of autism spectrum disorder using cry features”. This study developed a new method for distinguishing children with autism from typically developing children, based on two different features of their cries. The method involved training a model on cry features from one set of children, and then testing the model on a new set of cries from different children. The results (e.g., high sensitivity and specificity) are impressive, and the implications are intriguing. However, there are several details that are unclear and I have some concerns that might limit the applicability of this research, unless the authors can address them. These major concerns are described below, followed by some minor suggestions that will improve the readability of the manuscript.

**Major concerns:**

1. After reading the paper, I still do not have a clear understanding of the cry features that were useful for screening for ASD. As I understood it, MFCC and SONE are both related to power/amplitude/loudness of the cry, and the features that you used are based on the variance in MFCC and SONE from one temporal window to the next. So is this something that would be perceptible to a human ear? Is it a quality that can be described in terms of vocal quality (e.g. “shrillness”, “hoarseness”, “raspiness”)? Is the quality specific to cries, or would you also expect to see similar characteristics in the vocal quality of children who are talking rather than crying? You do not need to answer each of these specific questions, but if it is possible to describe the results or implications in layman’s terms, I think many readers would appreciate it. On the other hand, if it is something that only a computer can detect with no discernable difference to a human, then it might be informative to say that (e.g., a single sentence in the discussion).

2. If children were crying for different reasons, wouldn’t that affect cry quality? For example, do you know if a hunger cry has the same qualities as a cry of pain? Unless you can verify that children with ASD produce the characteristic cry qualities regardless of the reason for crying, then it would be important to know which type of cry would be most accurate when actually screening a child.

3. If children in the training sample produced different numbers of cries, wouldn’t the children with more cry instances be weighted more heavily in the model than children with fewer cry instances?

4. The children in the TD group were not assessed to verify that they were all truly typically developing. So this group could have included children with ASD that had not yet been diagnosed, or children with some other non-ASD disorder. This is not a fatal flaw, but it should be mentioned as a possible limitation. Specifically, it limits your ability to say whether this method distinguishes children with ASD from all other non-ASD children (i.e., including TD children and children with other disorders) or if it simply distinguishes all non-TD children from TD children.

5. Related to the last point, the children were not followed up later, so you do not know if some of the TD children were diagnosed with ASD (or some other disorder) at a later age. If any of the TD children were diagnosed at a later age, this would decrease the sensitivity. In general, the method seems to be only as sensitive as the person who was providing the evaluations (rather than more sensitive, so it can detect children with ASD *before* they have obvious symptoms).

6. The current sensitivity & specificity are impressive, but this may be because you included only TD and ASD groups. In real-world settings (e.g., pediatrician offices), this method would need to be able to classify other groups of children too, such as those with health issues (e.g., asthma) or other developmental disorders (e.g., apraxia). It is unclear how these other groups (especially groups with atypical speech, like apraxia of speech or a stutter) would be classified by this method. If these children were classified as “not TD” that would drastically reduce the specificity of this method as an ASD screener, but might make it useful as a general screener that would then lead to a more comprehensive evaluation.

7. The goal is to diagnose children earlier than is currently possible, but the model is trained on children that are approximately 3 years of age (which is around the current average age of diagnosis). It would be important to know how accurate this method would be if used at 12 months, 18 months, or 24 months. You have begun to address this by testing the model on 61 children between 10 and 18 months of age, but the real test of sensitivity would require you to evaluate these 61 children after 3 years of age, so that you could say that the one child that was detected was truly the only one that was not TD.

**Minor suggestions:**

Pg. 1 & Pg. 8: The abstract on page 1 is completely different from the abstract on page 8 of the PDF.

Line 21: “correspondign” is misspelled

Line 27: “feamale” is misspelled

Line 34: “ignited” is metaphorical. A literal word might be better, such as “inspired” or “instigated”

Line 35: instead of “many researches”, try “much research”

Line 38: I would change “about 3 years” to “over 3 years”, because the reported average age of 3.1 years was just for the most severe cases. Less severe cases of ASD (e.g., PDD-NOS and Asperger’s) were diagnosed at much later ages (3.9 and 7.2 years, respectively).

Line 53: “features differences” could be changed to “feature differences”

Lines 58-59 (also 73-74): This method is unfamiliar to me. I feel confident that I understand what you mean, but it might be useful to include a citation.

Lines 94-97: I find this section to be unnecessary. All of this is redundant with the sub-headings below. This might be helpful if the method was very long or complicated, but I think yours is fairly straightforward.

Line 103: “set” should be plural, “sets”

Line 104: instead of “only a model was trained for screening male subjects”, try “the model was only trained for screening male subjects”

Line 112: “psychologist” should be plural, “psychologists”

Line 113: “was” should be added, to make it “diagnosis of ASD was established”

Lines 123-132: some of the column headers are self-explanatory and do not need to be described in the text (e.g., subject IDs, ages). The only time you need to explain further is when there is information that is not explained in the table, such as the reason for two recording device types, or the 4 participants without GARS scores.

Line 129: Does “GRAS grades” mean “GARS scores”?

Line 149: “balancing on” should be “balancing of”

Line 162: “there were asked” should be “they were asked”

Line 174: Why were the uvular/guttural parts of the cries removed? Is there evidence that these parts are not informative? If so, you could cite that evidence. Otherwise, you could add “because we believed [whatever the reason was]”

Line 177: what does it mean “to be comprehended by our audition”?

Line 229: “till” should be “until”

Line 230: “is appeared” should just be “appeared”

Line 240: In my opinion, this whole “Feature extraction” section would have been helpful if it was presented before the section about classifying based on features.

Line 255-256: I do not understand why you modified the spectral flatness features. If this is common practice, you could either just say that or provide a citation. If it was your own decision, you could say why you decided that it was necessary.

Line 270: “omitted by” should be “omitted for”

Line 271: I think this should have been said earlier, such as after line 249. While I was reading lines 253-255, I had been wondering how long a frame was.

Line 279: “details)” has a parenthesis attached

Lines 280-281 (and line 284, as well as lines 300-301): These are technical terms that I am not familiar with. Would it be useful to provide a citation?

Line 336: what is meant by “the only method”?

Line 353: “was used to train” should be “were used to train”

Line 358: “subject” should be plural, “subjects”

Line 368: I am not certain, but I think “claimed” should be “claims”

Lines 372-373: I think this is good, if it is intended to screen for atypical development in general. But if this method is meant to be specific for ASD screening, then this would result in an overall decrease in specificity.

Lines 480 (and elsewhere in the Appendix): you used some technical terms but not citations are provided.

Reviewer #3: This manuscript addresses an important question: identifying children with ASD manifestations using biological information, specifically, auditory signals from crying. One compelling result is heightened variability in ASD cries relative to TD children. The interpretation (though now on P. 24 in the Appendix, and needs to be more fully elaborated and presented sooner in the ms) is that the findings are not due to cry per se, but due to the specific statistical features of the cry instances that capture heighted variability of cries. The authors propose that because the increased variability may be an enduring, fundamental feature of ASD, for this reason the pattern holds constant across age. I agree that heightened variability in biological signals is a clear and an emerging trend in ASD and I think that this manuscript does add important new information for our understanding of ASD in toddlers.

There are several major technical/scientific and conceptual concerns that should be addressed though. One example (there are many, see below) of a technical concern is a lack of uniform processing strategy for cry instances, resulting in a huge range of cry durations (1/3 to 3 seconds). In addition, this manuscript requires substantial re-drafting to incorporate more recent references and rationale for certain points made, and other places require additional justification and elaboration. The text is somewhat sloppy and not well-prepared (however, this issue could be related to the language use concern noted below). Limitations should be articulated.

Please have a colleague or a student who is a native British English or American English speaker carefully review and edit the entire manuscript, including captions and appendices. If this is not possible, please use an English-language editing fee-based service. The language use issues (e.g. awkward language and word use; grammar) are significant, numerous, and should be addressed adequately prior to re-submission.

Despite these points, I think that the work has important strengths. I also think that given the importance of question on early ASD detection, the authors may be given an opportunity to revise the ms, provided that they carefully consider and address all points raised (for any points not addressed, please provide suitable rebuttals).

abstract

-there are currently 2 versions of the Abstract text – one provided along via the submission system and a second version included as part of the manuscript file.

1) Please review language usage in both and 2) please choose one version (not clear which is the final version?)

“The approach has been tested on a dataset including 14 male and 7 female children with ASD and 14 male and 7 female TD children, between 18 to 53 months of age.”

-but the algorithm was developed with male subjects data only? Why? What is the result when you test on male-only subset?

Introduction

“On the other hand, it is shown that fMRI [10] or EEG [11] can give discriminative features helping to diagnose ASD at earlier ages.”

This statement is not accurate, as ref [10] predicted future ASD diagnoses made by traditional means (ADOS, ADI-R), by using rs-fMRI data from 6 mo. The diagnoses themselves are still performed using observational instruments. This approach is not the same as one to be used for “screening” for ASD symptoms in the absence of a diagnosis and also it is not the same as one to be used for using biobehavioral information in lieu of conventional diagnoses.

Please be careful in your writing and distinguish (1) work reporting results that predict future ASD diagnoses/cases, or (2) work that is able to detect concurrent cases of ASD (i.e. diagnoses are made at the same time as classification), or (3) work that aims to establish biological features of ASD that can be themselves be used for ASD diagnoses in lieu of traditional instruments.

Here it may be helpful to add ref Denisova & Zhao, 2017 who used movement data from rs-fMRI from 1 mo to predict future atypical developmental trajectories in general (this report provides the earliest age of detection using rs-fMRI data in an unconventional way).

Please also add more recent references from various groups who report ASD classification/ screening using items from the ADOS that is more recent: Kuepper et al., 2020; Abbas et al., 2020. Also add more references for younger ages since that is your target population.

“Furthermore, approaches which involves methods such as fMRI or EEG, are hard to be used on children, especially children with autism.”

This statement requires support from the literature – what makes them “hard” or unsuitable? (excessive head movements, etc. Please provide specific reasons and supporting references for each point made) (e.g. Denisova 2019).

Oller et al. 2010 paper should be discussed in the Introduction.

P. 2 Introduction

“To the best of our knowledge, there is only one work that proposed a method for identifying ASD children using only cry [24]. They used a dataset of 5 children with ASD and 4 TD children older than two years of age. They extracted 187 sound features from which 55 features were selected, using forward selection.”

This reference #24 seems to be work from the same group as authors of the current manuscript. Please refer to this reference appropriately (“We” instead of “They”, or “Work from our group has shown that …”)

Is there an overlap of children from study described in Reference #24 with the current work? Please clarify.

P. 3 Introduction

“It should be mentioned that the extracted features are age invariant and have been used to screen children with ASD between 18 to 53 months of age.”

This point about the current method’s age-invariance is emphasized repeatedly as being a strength of the proposed approach, but as written, it is unclear why or even if this might be the case.

It may be problematic in terms of specific behaviors not present at all ages. For an approach to be scientifically valid, it must have ecological validity with respect to natural human developmental behaviors.

For instance, you may encounter fewer crying instances at older ages, even if the older child is undergoing therapy (these children may adapt a different response instead of crying, and throw temper tantrums and present loud yelling and vocalizations).

The link to the fact that the statistical feature used (e.g. capturing heightened variability) is or could be feature of ASD that may be age-invariant is not really articulated, until the last page in the Appendix.

Crying (and voice/speech) is a motor act. Research from various groups has now established that infants, children, adults with ASD have problems in the motor domain and in coordination of motor domain with other modalities. Please cite references to this highly relevant work and please discuss in the Discussion.

P. 4 Subjects

Please review ascertainment of and description of ASD diagnoses. Please re-write this section in a more succinct, formal prose.

“The inclusion criteria of the ASD subjects were those who had been just diagnosed as autistic, based on DSM V . . .”

This statement is not correct.

-DSM-5 provides for diagnoses of Autism Spectrum Disorder only. “Autism” is not one of the diagnoses in DSM-5.

-also, please note: the correct name for the fifth DSM version is DSM-5, not “DSMV”.

-Please provide justification why ADOS (Toddler Module or Module 1), and/or ADI-R was not given. Is there a valid/official translation of the ADOS in Farsi that can be administered in Iran? Is there a research version available that you can request permission to translate from WPS, the publisher? If not, this information should be incorporated, as we clearly need version of ADOS (and/or ADI-R) for Islamic countries. However, as this is a limitation of the work, you should to provide a more thorough explanation of why the assessments are missing from your study and what are the plans to address this important issue in the future.

-I just checked for available official/published translations on the ADOS website and I did not see a Farsi/Persian version.

https://www.wpspublish.com/published-translations

-Right now from the text as written it is not clear why these instruments were not given to children in this study.

“For full descriptions of these procedures, see our previous work [27].”

Please do not refer the readers to these key details elsewhere. You may write this section in a more succinct manner but crucial information and justification must be given here in this ms, especially as there are no strict word limitations in PLOS One.

p. 6 Table 1.

Typically Developed –> “Typically Developing”. The name of this normal control children’s cohort should be revised to state “Typically Developing”.

“Subjects” – ‘participants’, ‘participating children’.

P. 7, end of the Subject section

This section is not well-prepared and some information (on the devices) is explained in the following section. Please re-draft and re-organize this material in these adjacent sections.

“Furthermore, the number of subjects with high-quality recordings was 6 ASD and 4 TD.”

This statement indicates that the main portion of the data was acquired with typical cell phones. Please re-run the main analysis, including only participants’ data obtained with cell phones.

P. 7, 8 Data collection and preprocessing

Need to provide more details about how the preprocessing step was implemented.

Raw data handling: need to clearly state which stages were automatic and which were manual.

Data samples: What are they being equated upon? Equal duration?

“The reason for using various devices was to have a device independent model.”

This statement holds if and only if you have already validated your algorithm on some standard data acquisition device (i.e. a high-quality recording device or a cell-phone with high quality audio recording). You need to be confident that the data are acquired in a robust and consistent way. After this step is demonstrated and findings are robust, then you may include a variety of recording devices. For this reason, it would be important to run an analysis using cell-phone data only, as cell-phone acquisitions represent the majority of your data.

P. 8.

“In this study, the final samples were between 320 milliseconds to 3 seconds.”

It is not acceptable to have such a wide range of data length (1/3 s to a full 3 s). Have you run an analysis that rules out data length as a contributing factor? What about the number of instances per each subject/participant? You need to have a similar number of instances across subjects.

Please provide clear justification for all analytic choices made. Please select one or more criteria, such as equal length of a cry instance, or the number of discrete crying instances, etc.

P. 9 Caption of Fig1:

“All instances of a subject have the same color.” – Do you mean to say: all instances *belonging* to a subject/participant?

P. 9 “An example of case 1 is tip toe walking in children with ASD, which is common in about 25% of these children who do it most of the times.”

-Please provide a reference from the literature for the tip toe walking example.

-“most of the times” – “most of the time”.

P. 13 Please review this technical description for mathematical accuracy:

“Frame-wise temporal derivative means subtracting the value of a frame from the value of the next frame.”

Subtracting value of one frame from the next frame produces a difference of values.

“Finally, to compute the features, each instance was divided into 20 milliseconds frames.”

What is the rationale for 20ms resampling / duration of each frame’s duration? Ideally you need to follow the Nyquist theorem (if you did, this needs be stated) for determining the proper sampling rate for a given phenomena of interest. However, you seem to have a fixed sampling rate of 44.1 kHz for all of the devices. All of this information needs to be reconciled in a technically and mathematically precise fashion and produce correct and concise descriptions for all methods.

“Also, we added a few other promising features.”

Please create a new Table summarizing fundamental descriptors and rationale for inclusion/exclusion of specific properties in this study.

P. 20 (2nd page of Discussion and conclusion)

“Finally, the proposed approach was also tested on 57 subjects, 56 TD children and one who diagnosed as Unspecified Neuro Developmental Disorder (UNDD), between10 to 18 months of age. The classifier distinguished the UNDD child from the rest. This suggests that a) the system can be used for children under 2 years of age, and b) it may be able to distinguish more general UNDD children from TD children.”

Please see P. 51 of DSM-5 on assigning DSM-IV diagnoses using DSM-5 format.

The Discussion section is the first time this 2nd dataset is mentioned (no mention in the methods, results, etc.). Please provide adequate details on this analysis in earlier sections as relevant.

Appendix.

This section contains key methods information necessary to understand analytical approach and is short enough to be in the main body of the ms. The ms is currently missing these details. Please move this information into the main body under the Methods section.

P. 23, 24, 25

Please address the following queries related to preprocessing of the data:

A1) Are these data continuous? i.e. have you retained original continuity of the data (i.e. as the data were acquired?). If not, how instances were strung together needs to be explained.

A2) Pipeline/preprocessing:

A 2a) How are technical issues of data preprocessing addressed here? (how are known problems addressed?)

A 2b) illustration of raw data of interest (cry instances of ASD and TD participants)

A 2c) equal length of samples/instances? All of this information is missing

A3a) What software/scripts was used to handle the raw data (.wav files)? (What is the software used for input/output? Open source?)

A3b) What software was used to program the classifiers (MATLAB, Python, etc.; which toolboxes or libraries were used if any?)

A4) Appendix – P. 24

“This feature supports our expert’s report saying variations in the cry of ASD children is more than TD children.”

A4a) Please present this report/ anecdotal variations in cry sooner in the ms.

A4b) This information touching upon variability parameters in ASD cry data is highly significant and should be elaborated upon in the Discussion. There are now many papers in the ASD field reporting that autism biological signals (from children and adults with ASD and work with infants at high risk for autism) are characterized by increased/heightened levels of variability (e.g. Denisova & Zhao, 2017).

Your result is in line with these other reports from many groups worldwide and uniquely adds to the growing research indicating that a marker of higher variability is an important feature in ASD. Perhaps that is why you find it age-invariant: it is not cry per se, but statistical features of the cry instances that hold constant across age. Please be sure to discuss this point in the Discussion.

6. PLOS authors have the option to publish the peer review history of their article (what does this mean?). If published, this will include your full peer review and any attached files.

Reviewer #1: No

Reviewer #2: No

Reviewer #3: No

---

## [Author Response · Author response to Decision Letter 0]

21 Jun 2020

The detailed response is given in the file prepared for it. We like to give our great appreciation to the reviewers for their constructive and important feedbacks. we are sure that the paper is in a much better shape now than before and the readers can benefit better from it.

Best regards,

---

## [Decision Letter · Decision Letter 1]

12 Aug 2020

PONE-D-19-32813R1

Early screening of autism spectrum disorder using cry features

PLOS ONE

Dear Dr. Moradi,

Thank you for submitting your manuscript to PLOS ONE. After careful consideration, we feel that it has merit but does not fully meet PLOS ONE’s publication criteria as it currently stands. Therefore, we invite you to submit a revised version of the manuscript that addresses the points raised during the review process.

We look forward to receiving your revised manuscript.

Kind regards,

Zhishun Wang, Ph.D.

Academic Editor

PLOS ONE

Reviewers' comments:

Reviewer's Responses to Questions

**Comments to the Author**

1. If the authors have adequately addressed your comments raised in a previous round of review and you feel that this manuscript is now acceptable for publication, you may indicate that here to bypass the “Comments to the Author” section, enter your conflict of interest statement in the “Confidential to Editor” section, and submit your "Accept" recommendation.

Reviewer #1: All comments have been addressed

Reviewer #3: (No Response)

2. Is the manuscript technically sound, and do the data support the conclusions?

Reviewer #1: Yes

Reviewer #3: Yes

3. Has the statistical analysis been performed appropriately and rigorously? 

Reviewer #1: Yes

Reviewer #3: Yes

4. Have the authors made all data underlying the findings in their manuscript fully available?

Reviewer #1: Yes

Reviewer #3: (No Response)

5. Is the manuscript presented in an intelligible fashion and written in standard English?

Reviewer #1: No

Reviewer #3: No

6. Review Comments to the Author

Reviewer #1: The authors have been very responsive to the last round of reviews and I'm happy with the manuscript as is; no suggested edits on my part

Reviewer #3: I thank the authors for carefully addressing previous points made. I am marking this as a major revision because I would like to see the authors carefully and thoroughly address the following new and remaining points.

Abstract in the current revised version

“Due to the importance of automatic and early autism screening, in this paper, a cry- based screening approach for children with Autism Spectrum Disorder (ASD) is introduced. During the study, we realized that the ASD specific features are not necessarily observable among all children with ASD and among all instances of each child. Therefore, we proposed a new classification approach to be able to find such features and their corresponding instances. We tested the proposed approach and found two features that can be used to distinguish groups of children with ASD from Typically Developing (TD) children. In other words, these features are present in subsets of children with ASD not all of them. The approach has been tested on a dataset including 14 boys and 7 girls with ASD and 14 TD boys and 7 TD girls, between 18 to 53 months old. The sensitivity, specificity, and precision of the proposed approach for boys were 85.71%, 100%, and 92.85%, respectively. These measures were 71.42%, 100%, and 85.71% for girls, respectively. “

1) The abstract, the final version of which was not available in the previous submission, requires substantial changes and must be fully re-written to include rationale for the current work. More details are needed about the analytic methods (including acquisition methods) and conclusion. First of all, the implications/conclusion of the work is completely missing and must be added. Overall, it should reflect all of the changes and edits in the current ms.

2) “Due to the importance of automatic and early autism screening” Please consider providing additional rationale for the importance of early ASD screening. It is mentioned in passing in the ms on P. 1 but I would like to see some more elaboration for why it is important to detect ASD (or ASD signs) in the main body of the ms /in the Introduction.

3) There remain many instances of awkward language use, which often leads to confusion about the intended meaning. I would like to see a version of this ms after appropriate language edits have been implemented, because I do not think that the Authors have adequately dealt with the language issues in this version. I would suggest the use of a professional editing service, or have a colleague who is a native British English or American English speaker help edit the ms.

4) Some examples of awkward language use, but there are others. Also please be very clear about the intended meaning. There is confusion about screening for ASD manifestations in children below 18 months *who have not yet been diagnosed with an ASD* and confirming ASD in children below 18 mo – please clarify what is meant by using clear language.

“53 As mentioned above, there are studies tried to screen children with ASD under 18 months,”

Should read: “As mentioned above, multiple studies attempted to screen children for ASD below 18 months of age”

“widespread expertness for autism diagnosis“

Should read: “expertise for diagnosing autism”

5) Some references are missing in the ms – For example, it does not seem that this reference is in the main text.

Paliwal KK, Lyons JG, Wójcicki KK, editors. Preference for 20-40 ms window duration in speech analysis. 2010 4th International Conference on Signal Processing and Communication Systems; 2010: IEEE.

6) The fact that ADOS is not available at all in Farsi is still not mentioned in the body of the ms. I think it is important to mention this fact, in case some readers wonder why you did not administer the ADOS to your participants. Please address in the main body of the ms why participants were not administered the ADOS.

7. PLOS authors have the option to publish the peer review history of their article (what does this mean?). If published, this will include your full peer review and any attached files.

Reviewer #1: No

Reviewer #3: No

---

## [Author Response · Author response to Decision Letter 1]

3 Oct 2020

The comments by the 3rd reviewer, especially the issue regarding using a professional editing service, has been addressed. The response to reviewers document includes all the corrections/comments regarding the reviewer's comments.

---

## [Decision Letter · Decision Letter 2]

20 Oct 2020

Early screening of autism spectrum disorder using cry features

PONE-D-19-32813R2

Dear Dr. Moradi,

We’re pleased to inform you that your manuscript has been judged scientifically suitable for publication and will be formally accepted for publication once it meets all outstanding technical requirements.

Kind regards,

Zhishun Wang, Ph.D.

Academic Editor

PLOS ONE

Additional Editor Comments (optional):

Reviewers' comments:

Reviewer's Responses to Questions

**Comments to the Author**

1. If the authors have adequately addressed your comments raised in a previous round of review and you feel that this manuscript is now acceptable for publication, you may indicate that here to bypass the “Comments to the Author” section, enter your conflict of interest statement in the “Confidential to Editor” section, and submit your "Accept" recommendation.

Reviewer #1: All comments have been addressed

Reviewer #3: All comments have been addressed

2. Is the manuscript technically sound, and do the data support the conclusions?

Reviewer #1: Yes

Reviewer #3: Yes

3. Has the statistical analysis been performed appropriately and rigorously? 

Reviewer #1: I Don't Know

Reviewer #3: Yes

4. Have the authors made all data underlying the findings in their manuscript fully available?

Reviewer #1: Yes

Reviewer #3: (No Response)

5. Is the manuscript presented in an intelligible fashion and written in standard English?

Reviewer #1: Yes

Reviewer #3: Yes

6. Review Comments to the Author

Reviewer #1: (No Response)

Reviewer #3: (No Response)

7. PLOS authors have the option to publish the peer review history of their article (what does this mean?). If published, this will include your full peer review and any attached files.

Reviewer #1: No

Reviewer #3: No

---

## [Editor Report · Acceptance letter]

2 Dec 2020

PONE-D-19-32813R2 

Early screening of autism spectrum disorder using cry features 

Dear Dr. Moradi:

I'm pleased to inform you that your manuscript has been deemed suitable for publication in PLOS ONE. Congratulations! Your manuscript is now with our production department. 

Kind regards, 

on behalf of

Dr. Zhishun Wang 

Academic Editor

PLOS ONE